# Pancreatic Cancer and Microenvironments: Implications of Anesthesia

**DOI:** 10.3390/cancers14112684

**Published:** 2022-05-28

**Authors:** Hou-Chuan Lai, Yi-Wei Kuo, Yi-Hsuan Huang, Shun-Ming Chan, Kuang-I Cheng, Zhi-Fu Wu

**Affiliations:** 1Department of Anesthesiology, Tri-Service General Hospital and National Defense Medical Center, Taipei 114, Taiwan; doc31454@mail.ndmctsgh.edu.tw (H.-C.L.); yixiun@mail.ndmctsgh.edu.tw (Y.-H.H.); whcken@mail.ndmctsgh.edu.tw (S.-M.C.); 2Department of Anesthesiology, Kaohsiung Medical University Hospital, Kaohsiung Medical University, Kaohsiung 807, Taiwan; 900121@kmuh.org.tw (Y.-W.K.); kuaich@kmu.edu.tw (K.-I.C.); 3Department of Anesthesiology, Faculty of Medicine, College of Medicine, Kaohsiung Medical University, Kaohsiung 807, Taiwan; 4Center for Regional Anesthesia and Pain Medicine, Wan Fang Hospital, Taipei Medical University, Taipei 116, Taiwan

**Keywords:** pancreatic cancer, tumor microenvironment, anesthesia

## Abstract

**Simple Summary:**

Pancreatic cancer is a lethal malignant neoplasm with less than 10% 5-year relative survival after the initial diagnosis. Several factors may be related to the poor prognosis of pancreatic cancer, including the rapid tumor progression, increased metastatic propensity, insignificant symptoms, shortage of early diagnostic biomarkers, and its tendency toward resistance to both chemotherapy and radiotherapy. Pancreatic neoplastic cells interact intimately with a complicated microenvironment that can foster drug resistance, metastasis, or relapse in pancreatic cancer. In addition, evidence shows that perioperative factors, including surgical manipulation, anesthetics, or analgesics, might alter the tumor microenvironment and cancer progression. This review outlines the up-to-date knowledge of anesthesia implications in the pancreatic microenvironment and provides future anesthetic strategies for improving pancreatic cancer survival.

**Abstract:**

Pancreatic malignancy is a lethal neoplasm, as well as one of the leading causes of cancer-associated mortality, having a 5-year overall survival rate of less than 10%. The average life expectancy of patients with advanced pancreatic cancer does not exceed six months. Although surgical excision is a favorable modality for long-term survival of pancreatic neoplasm, metastasis is initially identified in nearly 80% of the patients by the time of diagnosis, making the development of therapeutic policy for pancreatic cancer extremely daunting. Emerging evidence shows that pancreatic neoplastic cells interact intimately with a complicated microenvironment that can foster drug resistance, metastasis, or relapse in pancreatic cancer. As a result, the necessity of gaining further insight should be focused on the pancreatic microenvironment contributing to cancer progression. Numerous evidence reveals that perioperative factors, including surgical manipulation and anesthetics (e.g., propofol, volatile anesthetics, local anesthetics, epidural anesthesia/analgesia, midazolam), analgesics (e.g., opioids, non-steroidal anti-inflammatory drugs, tramadol), and anesthetic adjuvants (such as ketamine and dexmedetomidine), might alter the tumor microenvironment and cancer progression by affecting perioperative inflammatory or immune responses during cancer surgery. Therefore, the anesthesiologist plays an important role in perioperative management and may affect surgical outcomes. However, the literature on the impact of anesthesia on the pancreatic cancer microenvironment and progression is limited. This review summarizes the current knowledge of the implications of anesthesia in the pancreatic microenvironment and provides future anesthetic strategies for improving pancreatic cancer survival rates.

## 1. Introduction

Pancreatic cancer (PC) is a lethal malignant neoplasm. It is considered one of the leading causes of cancer-associated mortality, having a 5-year overall survival rate of less than 10% [1]. The average life expectancy of patients with advanced PC does not exceed six months [2]. Several factors related to the poor prognosis of PC include insignificant symptoms, shortage of early diagnostic biomarkers, rapid progression of the disease, increased metastatic propensity, and the disease’s tendency toward resistance to both chemotherapy and radiotherapy [3,4]. Early resection of the neoplasm is potentially the most successful treatment regime for such aggressive malignancy. However, by the time of diagnosis, metastasis is initially identified in approximately 80% of the patients, making PC treatment extremely challenging [5,6]. Despite several studies outlining various signal pathways involved in PC progression, the mechanisms of tumor progression are poorly understood.

The pancreatic microenvironment plays an important role in tumorigenesis [7]. As cancer progresses, alterations of surrounding tissue stroma develop rapidly. A key role of any non-transformed tissue stroma is to maintain a homeostatic response to shelter the immune system, the vascular system, and connective tissue components. Cancer hijacks the physiological responses to build a tumor microenvironment beneficial for cancer progression [7]. The pancreatic microenvironment includes the surrounding desmoplastic stroma, such as the epithelial-mesenchymal transition (EMT), and immunosuppressive pathways [7]. Evidence reveals that anesthetics or analgesics might alter the tumor microenvironment and the progression of the cancer by affecting perioperative inflammatory or immune responses during PC surgery [8,9]. Soliz et al. reported that propofol anesthesia was associated with no or low-grade complication compared with desflurane anesthesia in PC surgery [10]. In addition, in a retrospective study, we found that desflurane anesthesia was associated with poorer survival than propofol anesthesia in PC surgery [11]. Zhang et al. demonstrated that intraoperative intravenous lidocaine infusion increased survival in PC surgery [12]. Therefore, anesthesiologists may improve clinical outcomes by using preferential anesthetics. This review summarizes an overview of current knowledge of implications of anesthesia in the pancreatic microenvironment, and provides future anesthetic strategies for improving PC survival.

## 2. Patient Factors: Hyperglycemia and Obesity

### 2.1. Hyperglycemia

Diabetes mellitus (DM) and PC are intimately related, as high blood glucose levels promote PC proliferation, invasion, EMT, and metastasis [13,14]. In addition, insulin resistance, hyperinsulinemia, hyperglycemia, and chronic inflammation are the mechanisms of type-2-DM-associated PC [15].

#### 2.1.1. Laboratory Studies

Recently, Otto et al. attributed a role to the type-2-DM-related hyperglycemic inflammatory micromilieu in the acquisition of malignancy-associated alterations in premalignant pancreatic ductal epithelial cells, thus providing new insights into how hyperglycemia might promote PC initiation [16]. It is well-known that EMT of pancreatic ductal epithelial cells develops in correlation with hyperglycemia or macrophages [17,18]. Moreover, hyperglycemia aggravates microenvironment hypoxia, accelerates EMT, and then promotes the metastatic ability of PC. PC is generally hypoxic due to its avascular morphology, and PC cells express high levels of HIF-1α and MMP-9 for promoting tumor growth, invasion and metastasis in a hypoxic environment [19]. In addition, the accumulation of HIF-1α induced by hyperglycemia might promote pancreatic glycolysis to facilitate cancer progression [20]. Zhou et al. reported that the high-glucose microenvironment accelerated PC growth [21]. With regard to the VAs, Guo et al. reported that isoflurane promoted glucose metabolism through upregulation of *miR-21* and suppressed mitochondrial oxidative phosphorylation in ovarian cancer cells [22]. Dong et al. reported that dezocine, an opioid analgesic, promoted glucose metabolism and impaired the proliferation of lung cancer cells [23]. However, Codd et al. reported that opioid agonists did not elevate blood glucose and lacked an insulin-reducing effect [24]. Han et al. reported that indometacin, an inhibitor of cyclooxygenase (COX)-2, ameliorated high-glucose-induced proliferation and invasion via upregulation of E-cadherin in PC cells [25]. Current laboratory data on the effect of anesthesia on glucose metabolism in PC are limited, and further investigation is required.

#### 2.1.2. Clinical Studies

Insulin resistance, hyperinsulinemia, hyperglycemia, and chronic inflammation are the mechanisms of type-2-DM-associated PC [15]. Recently, type 2 DM was shown to reduce the likelihood of cancer survival, and was significantly correlated with comorbidity and poor prognosis in patients undergoing PC surgery [15]. In addition, metformin may lower the probability of PC. By contrast, insulin therapy may amplify the probability of PC [15]. In another study, approximately 85% of PC patients exhibited impaired glucose tolerance associated with DM and had a reduced overall survival rate [26]. Elderly patients with new-onset DM are at higher risk of developing PC than the general population [26]. Therefore, new-onset DM and hyperglycemia serve as important screening tools to diagnose asymptomatic PC and improve PC survival [26]. Sandini et al. reported that preoperative blood glucose ≥ 140 mg/dL was associated with poor long-term outcomes in patients undergoing resection for PC [27]. Conti et al. reported that anti-diabetic drugs represented a significant protective factor against mortality among older adults with metastatic PC [28]. However, in a recent meta-analysis study, blood glucose, fasting blood glucose, and glycated hemoglobin (HbA1c) levels were not associated with the survival of patients with PC [29]. 

Liu et al. reported that the blood glucose levels of the DM patients in the propofol group were significantly lower than those in the sevoflurane group during gastric cancer surgery. This result indicated that the effect of propofol on glucose metabolism under surgical stimulation was less than that of sevoflurane [30]. Epidural blockade with bupivacaine attenuated the hyperglycemic response to surgery by modifying glucose production in colorectal surgery [31]. Current clinical data on the effect of anesthesia on glucose metabolism in PC are limited. Further investigation is required to determine the effects of anesthetics and analgesics on glucose metabolism in PC (Figure 1). 

### 2.2. Obesity

Obesity-associated adipose tissue inflammation may play a central role in the development of PC and the promotion of PC growth [32]. Chronic inflammation, hormonal effects, circulating adipokines, and adipocyte-mediated inflammatory and immunosuppressive microenvironments are involved in the association of obesity with PC [33]. The tumor-promoting effects of obesity occur at the local level via adipose inflammation and associated alterations in the microenvironment, as well as systemically via circulating metabolic and inflammatory mediators associated with adipose inflammation [34]. 

In a review article, Heil et al. reported that anesthetics with the effect of inhibiting obesity-induced inflammation may improve postoperative outcomes [35]. Eley et al. concluded that VAs, ketamine, opioids, propofol, and regional anesthesia have been shown to modulate parts of the immune system in patients with obesity [36]. 

#### 2.2.1. Laboratory Study

Incio et al. reported that obesity-induced inflammation and desmoplasia promoted PC progression and resistance to chemotherapy [37]. Until now, there have been no laboratory studies on the effects of anesthetics in obesity-induced inflammation and PC progression, and further investigation is necessary.

#### 2.2.2. Clinical Studies

Recently, Zorbas et al. showed that obesity was significantly associated with higher risk of postoperative complications and mortality in patients with body mass index ≥ 40 after pancreatoduodenectomy [38]. Li et al. reported that the lean body-weight-based dosing of propofol had more potent antioxidant and anti-inflammatory effects on morbidly obese patients than the total body-weight-based dosing during anesthesia induction [39]. Until now, there have been no clinical studies on the effects of anesthetics in obesity-induced inflammation and PC progression, and further investigation is necessary. 

## 3. Tumor Factors: EMT, Hypoxia-Inducible Factor-1α (HIF-1α), Matrix Metalloproteinases (MMP)-9 Expression, Inflammation, Apoptosis, Autophagy, and Oxidative Stress

### 3.1. EMT

The development of EMT originates in the conversion of epithelial cells to motile mesenchymal stem cells [40], which is based on many essential processes involving embryonic progression, tissue formation/fibrosis, and wound repairing [40]. Moreover, the initiation of EMT contributes to tumor growth, therapy resistance, and tumor spreading [40]. In the case of high EMT expression in tumors, deterioration of overall outcomes and metastases is inevitable. [40,41,42,43]. However, research on the direct effects of specific anesthetics on EMT of PC is currently lacking, and further investigation is required. 

#### 3.1.1. Laboratory Studies

Anesthesia and analgesia may affect EMT [25,44,45,46,47,48,49,50,51,52,53,54,55,56,57,58,59,60,61]. Studies have reported that propofol suppressed EMT in esophageal cancer, choriocarcinoma, breast cancer, thyroid cancer, lung cancer, gastric cancer, hepatocellular carcinoma (HCC), renal cell carcinoma (RCC), prostate cancer, and oral squamous cell carcinoma cells [45,46,47,48,49,50,51,52,53,54]. By contrast, Ren et al. reported that desflurane induced EMT and metastasis in colorectal cancer through deregulation of the miR-34a/LOXL3 axis [44]. Opioids promoted EMT in breast and lung cancers via mu- or delta-opioid receptors [55,56]. Zhang et al. showed morphine-induced EMT in esophageal carcinoma cells [57]. However, sufentanil inhibited EMT by acting on NF-κB and Snail signaling pathways to inhibit proliferation and metastasis of esophageal cancer [58]. Lidocaine suppressed EMT in ovarian cancer cells [59]. However, high concentrations of levobupivacaine significantly increased EMT in the A549 lung cancer cell line, and enhanced metastasis in mice [60]. COX-2 inhibitors may suppress EMT in oral squamous cell carcinoma [61]. Han et al. reported that indometacin reduced the expression levels of MMP-2, MMP-9, and vascular endothelial growth factor (VEGF) by upregulation of E-cadherin, inhibiting proliferation and invasion of PC [25]. Zheng et al. observed the benefit of EMT inhibition due to the use of chemotherapy in PC treatment [62]. To the best of our knowledge, NSAIDs may inhibit EMT expression in PC. Propofol may inhibit EMT, but VAs may promote EMT in different cancers. On the other hand, opioids and LAs may induce uncertain effects (both positive and negative) on EMT. Laboratory research on the direct effects of specific anesthetics on EMT in PC is currently lacking. Further investigation is needed (see existing studies in Table 1 and Figure 1).

#### 3.1.2. Clinical Studies

Clinical research on the direct effects of specific anesthetics on the EMT of PC is currently lacking. Further investigation is urgently required (Table 1 and Figure 1).

### 3.2. HIF-1α

A review article showed that HIF-1α expression enhanced PC cell proliferation through multiple mechanisms by inducing neoplastic features and mediating tumorigenic crosstalk between tumor and stromal cells [91]. 

#### 3.2.1. Laboratory Studies

Propofol could attenuate PC cells’ malignant potential by inhibiting HIF-1α and VEGF expression [73]. VAs enhance angiogenesis through HIF-1α activity in prostate and lung cancers [92]. Isoflurane upregulated the levels of HIF-1α and exerted a protumorigenic effect on a human RCC cell line [93]. However, in the neuroglioma cell line, sevoflurane decreased HIF-1α expression via *miR-210*, while desflurane downregulated HIF1-α and MMP-9 expressions via *miR-138* and *miR-335*, respectively [94]. Opioids were shown to promote tumor angiogenesis in a breast cancer cell by stimulation of δ-opioid receptors in breast cancer cells, leading to activation of HIF-1α and expression of COX-2 via PI3K/Akt stimulation [95]. However, Koodie et al. reported that morphine suppressed tumor angiogenesis by inhibiting HIF-1α expression in mouse Lewis lung carcinoma cells [96]. Okamoto et al. showed that HIF-1α activation conferred resistance to lidocaine-induced cell death in the RCC cell line [97]. Zhou et al. revealed that inhibition of HIF-1α by meloxicam (a selective COX-2 inhibitor) could suppress angiogenesis and enhance apoptosis of HCC cells [98]. To the best of our knowledge, propofol may reduce HIF-1α expression in PC. Based on the limited data, further investigation is required and encouraged to determine the effects of VAs, LAs, and NSAIDs on HIF-1α expression in PC (Table 1 and Figure 1). 

Recently, Yue et al. demonstrated that HIF-1α positively regulated miR-212 expression and resulted in pancreatic ductal adenocarcinoma progression [99]. Propofol inhibited ovarian cancer cells growth and glycolysis by elevating miR-212-5p expression [100]. Higher miR-212-5p expression showed a neuroprotective effect in rats with isoflurane-induced cognitive dysfunction by inhibiting neuroinflammation [101]. He et al. reported that δ-opioid receptor activation modified miR-212 expression in the rat kidney under prolonged hypoxia [102]. Until now, there have been no laboratory studies on the effects of anesthetics on miR-212 expression and PC progression; further investigation is necessary.

#### 3.2.2. Clinical Studies

In a systemic review and meta-analysis, Raji et al. found that miR-212 could be a novel potential biomarker in cancer diagnosis and prognosis [103]. High levels of miR-212 indicated poor prognosis in PC, and low levels of miR-212 indicated poor prognosis in other cancers [103]. Until now, there have been no clinical studies on the effects of anesthetics on HIF-1α or miR-212 expression and PC progression; further investigation is necessary.

### 3.3. MMP-9

MMPs are part of the zinc-dependent proteolytic metalloenzyme family that may play a role in the early diagnosis and prognosis of PC. The higher expression of particular MMPs may also correlate with metastatic disease and/or poorer prognosis [104,105,106,107]. MMP-9, well-known as one of the most investigated MMPs, corrupts the extracellular matrix components, resulting in pathophysiologic alterations [108]. Impairment of MMP-9 expression and regulation affects various dysfunctions, including tumorigenesis, and MMP-9 suppression can be targeted in anticancer therapeutics [108]. Anesthesia may affect MMP-9 expression [74,94,109,110,111,112,113,114,115].

#### 3.3.1. Laboratory Studies

Yu et al. reported that propofol inhibits PC growth by suppressing MMP-9 expression [74]. Sevoflurane and desflurane inhibited glioma cell proliferation and migration via downregulation of MMP-9 [94]. Sevoflurane and desflurane reduced the invasion of colorectal and neuroglioma cancer cells through downregulation of MMP-9 [94,110]. Moreover, sevoflurane inhibited the proliferation and invasion of HCC cells through downregulation of MMP-9 [111]. Zhang et al. showed that fentanyl inhibited tumor growth and cell invasion in colorectal cancer by downregulation of miR-182 and MMP-9 expression [112]. In addition, the antitumor effects of morphine are associated with a reduction in the level of MMP-9 [113]. Both lidocaine and ropivacaine inhibited TNFα-induced invasion of lung adenocarcinoma cells in vitro by blocking MMP-9 expression [114]. Based on the limited data, further investigation is needed to clarify the effects of anesthetics and analgesics on MMP-9 expression in PC (Table 1 and Figure 1).

#### 3.3.2. Clinical Studies

Wang et al. reported that MMP-9 in the propofol group was significantly lower than in the sevoflurane group in lung cancer patients who received surgery [109]. In breast cancer patients, Kashefi et al. reported that novel NSAIDs may reduce MMP-2 and MMP-9 expression, which promotes angiogenesis and metastasis [115]. In summary, propofol may reduce MMP-9 expression in PC. Based on the limited data, further investigation is required to determine the effects of anesthetics and analgesics on MMP-9 expression in PC (Table 1 and Figure 1). 

### 3.4. Inflammation, and the Immune System

Inflammation, apoptosis, and autophagy can provide cellular defense, and impairments of these processes (rendering them deficient or overactivated) lead to pathological effects. Inflammation induces secretion of various cytokines and chemokines, and recruits various immune cells in reaction to oxidative stress or infection sites. Reflexively, enhancement of reactive oxygen species (ROS)-generation via inflammatory immune cells provokes oxidative stress and tissue injury. In addition, chronic inflammation not only produces high numbers of inflammatory mediators but also gives rise to oxidative stress [116]. Inflammatory processes have emerged as key elements in PC development and progression [117,118]. The relationship of chronic inflammation and cancer, as revealed in the pioneering work of Rudolf Virchow, has been observed for more than 150 years, especially in PC progression [118]. However, even in malignancy without preceding inflammation, cancer-induced inflammation, secretions of inflammatory factors, and immune cell infiltration are main characters in tumor initiation and advanced metastasis [118]. Anesthesia and analgesia may impact cellular immunity and inflammation during surgery, and thus affect cancer outcomes [119,120,121,122,123,124,125].

#### 3.4.1. Laboratory Studies

A recent laboratory study has shown that propofol enhances anti-inflammatory reactions and stimulatory effects on immune responses, which may be a potential benefit in the prevention of tumor recurrence. However, clinical evidence of the tumor suppression effects is inconclusive. [126]. Opioids influence the nervous system indirectly, as well as release biological amines that potentially impair innate immunity by suppressing natural killer (NK) cell cytotoxicity. [127]. However, a mu-opioid receptor (MOR) partial agonist, buprenorphine, intercepted the inhibition of NK cell cytotoxicity and progression caused by surgery in a rat mammary adenocarcinoma cell line [128]. Additionally, neoplasm is related to inflammation, and anti-inflammatory properties are identified in LAs. LAs may be able to reduce metastasis risk, but the molecular mechanism is not fully understood. [129]. With regard to NSAIDs, aspirin was associated with a decreased expression of markers for progression, inflammation, and desmoplasia in PC cell lines [82]. NSAIDs also reduced inflammation and induced apoptosis in rat osteosarcoma cells in vitro [130]. Thus, NSAIDs may attenuate inflammation in PC. Further laboratory research is necessary (see extant research in Table 1 and Figure 1). 

#### 3.4.2. Clinical Studies

The surgical treatment of PC is complicated by the prolonged nature of the surgery, the magnitude of the surgical stress, inflammatory response, immunosuppression, anesthesia-/epidural-induced hypotension, and blood loss, all of which cause oxidative stress [92,131]. In a retrospective study based on clinical pathological analysis, Huang et al. showed that the survival probability was reduced in patients with TNM stage III to IV, lymph node metastasis, higher CD4^+^IL-17^+^ level, and lower CD8^+^ expression, which implied that the tumor immune microenvironment may affect the outcome of PC [132]. Recently, Li et al. reported that high systemic immune-inflammation index levels were regarded as negative with regard to PC overall survival and cancer-specific survival [133]. Yamaguchi et al. demonstrated that propofol reduced the number of CD8^+^ T cells, whereas sevoflurane augmented the percentage of regulatory T cells in lung-cancer surgery patients [121]. Sevoflurane was revealed to devastate multiple pulmonary functions by releasing a series of inflammatory secretions in lung cancer patients undergoing perioperative one-lung ventilation [134]. However, another clinical study reported that sevoflurane inhibited pulmonary inflammatory cytokines [135]. Propofol combined with epidural anesthesia and epidural analgesia demonstrated less interference with the immune system (compared to propofol with intravenous analgesia) and led to fast recovery in patients undergoing radical resection of pulmonary carcinoma [125]. However, in a clinical study, Fant et al. reported that thoracic epidural analgesia with bupivacaine inhibits the neurohormonal but not the acute inflammatory stress response after radical retropubic prostatectomy [136]. Based on the published data, further investigation is required to determine the effects of anesthetics and analgesics on inflammation and cellular immunity in PC progression (Table 1 and Figure 1). 

### 3.5. Apoptosis

Apoptosis, also known as programmed cell death, is an important mechanism in maintaining the stability of the tissue environment. It is activated by various vital signal transduction cascades [137]. Recent studies reported that autophagy and apoptosis are key mechanisms to fight PC [137,138].

#### 3.5.1. Laboratory Studies

Evidence has shown that anesthesia may affect apoptosis [75,76,86,87,139,140,141,142,143]. Wang et al. reported that propofol inhibited migration, induced apoptosis of PC cells, and inhibited cell migration in PC cells in vitro by promoting *miR-34a*-dependent LOC 285194 and E-cadherin upregulation [75]. Propofol also induced apoptosis and increased gemcitabine sensitivity in PC cells via the nuclear factor-κB signaling pathway [76]. Wei et al. reported that isoflurane suppressed proliferation and increased apoptosis and autophagy by activating the AMPK/mTOR pathway in cervical carcinoma, both in vitro and in vivo [139]. In contrast, promotion of bladder tumor cell proliferation, invasion, and migration was reported to be induced by isoflurane instead of apoptosis inhibition [141]. Sevoflurane can suppress colon cancer cell proliferation and invasion, provoke apoptosis and autophagy, and regulate EMT, actions that may result from the sevoflurane-induced inhibition of the ERK signaling pathway [140]. Celik et al. showed that fentanyl prescription reduced the numbers of tumor cells and tumor stem cells, diminished the stem cell gene expression of markers, and enhanced apoptosis-related gene expression in PC cells [86]. However, the fact that opioids increased PC cell death was not entirely due to alterations in apoptotic or necrotic pathways [142]. Bundscherer et al. showed that ropivacaine, bupivacaine, and sufentanil did not change the PC apoptosis rate and cell cycle distribution in clinical concentration [87]. Lin et al. showed that aspirin inhibited the proliferation and promoted the apoptosis of PC cells [143]. In summary, propofol and NSAIDs may induce apoptosis. However, VAs, opioids, and LAs may induce uncertain effects on apoptosis in PC. Based on the published laboratory data, further investigation is required to determine the effects of anesthetics and analgesics on apoptosis and progression in PC. 

#### 3.5.2. Clinical Studies

To the best of our knowledge, clinical research on anesthetics/analgesics and apoptosis in PC progression is still lacking; further research is needed.

### 3.6. Autophagy

Autophagy occurs in response to various stresses, inducing a series of intracellular degradation processes that involve presentations of organelle damage, abnormal protein expression, and nutrient deprivation [144]. PC cells respond to metabolic stress, nutrient starvation, hypoxia, and chemotherapy with the aid of autophagy. Consequently, autophagy repression may impede PC tumorigenesis [117]. In addition, inflammation and autophagy dysregulation are prevailing features of pancreatitis and PC [117]. Recently, Luo et al. reported that tendencies in PC toward invasion and resistance to standardized therapy derive from the obstruction of autophagy and apoptosis [137]. Therefore, it can be concluded that autophagy exerts biphasic effects on PC inhibition and facilitation [117,144]. 

#### 3.6.1. Laboratory Studies

Tumor cells depend on autophagy for tumor survival to a greater degree than normal cells, relying on it for their fast growth rate, for metabolic alteration, and to adapt to nutrient deprivation [145]. Autophagy enables cancer cells to respond to changes in nutrition supply by degenerating and recycling non-essential intracellular contents. Restriction of autophagy coupled with nutrient deprivation is an effective therapeutic strategy for cancer [146]. In addition, autophagy may improve cancer cell growth and survival in an unfavorable environment, such as that induced by the administration of cytostatic drugs; however, autophagy also constitutes an alternative mode of cell death [147]. Wang et al. reported that autophagy inhibitors combined with chemotherapeutic drugs showed promising results in PC treatments. Thus, autophagy may play a vital role in inhibiting PC progression [148].

Recent research shows that anesthesia may affect autophagy [140,149,150,151,152,153,154]. Wang et al. reported that propofol inhibited the proliferation, apoptosis, and cell cycle of HCC by inducing autophagy in vivo and in vitro [149]. Chen et al. showed that propofol exerted antitumor effects in cervical carcinoma cells by promoting autophagosome accumulation via inhibition of autophagy flux [150]. By contrast, sevoflurane-induced apoptosis and autophagy inhibited the proliferation and invasion of colon cancer cells [140]. Jiang et al. demonstrated that sufentanil impaired autophagic degradation and inhibited cell migration in lung cancer cells in vitro [151]. However, fentanyl induced autophagy via activation of the ROS/MAPK pathway, and reduced the sensitivity of cisplatin in lung cancer cells [152]. Izdebska et al. reported that lidocaine induced protective autophagy and cell cycle arrest in rat glioma cell lines [147]. Ropivacaine and ropivacaine-loaded liposomes impaired autophagy and suppressed cancer growth in B16 mouse melanoma lines and human cervical cancer cell lines [146]. Aspirin inhibited mTOR signaling, activated AMPK, and induced autophagy in colorectal cancer cells [153]. Palumbo et al. also reported that COX-2 inhibitor induced autophagy and reduced growth rate in human glioblastoma cell lines [154]. Laboratory research on the effects of anesthetics and analgesics on PC autophagy is lacking. Further investigation is required and encouraged. 

#### 3.6.2. Clinical Studies

Until now, clinical research on the effects of anesthetics/analgesics on PC autophagy is lacking. Further investigation is needed.

### 3.7. Oxidative Stress

Oxidative damage is induced through the excessive production and accumulation of free radicals, such as ROS, in mitochondria, leading to the impairment of lipids, proteins, and DNA [155]. Furthermore, fatty-acid peroxidation is influenced by oxidative stress, accompanied by mutagenic metabolites [155]. In addition, ROS and pro-inflammatory cytokines are considered to be important factors in the pathogenesis of PC [156]. Oxidative stress is involved in aging, inflammation, cancer, degenerative diseases, and can be induced by exposure to xenobiotics and drugs, such as anesthetics [131,157]. With the accumulation of evidence indicating a correlation between increased oxidative stress and worse postoperative outcomes, it is very important to choose anesthetics that minimize oxidative stress [131,157]. Anesthesiologists may improve clinical outcomes and reduce postoperative complications by selecting anesthetics that reduce perioperative oxidative stress. 

#### 3.7.1. Laboratory Studies

Zhang et al. demonstrated that propofol reduced oxidative stress in the human neuroblastoma cell line [158]. With regard to VAs, Wei et al. showed that oxidative stress was triggered by isoflurane, which elevated ROS levels and lowered the ratio of NAD+/NADH and ATP levels in cervical cancer cells [139]. In addition, there have been some indications in animal studies that sevoflurane may be neurotoxic in the developing brain, in part due to increased oxidative stress [159,160,161]. Regarding opioids, Almeida et al. stated that oxidative stress was ameliorated in glioma cell lines following treatment with therapeutic concentrations of morphine [162]. In contrast, Zhang et al. showed that morphine induced activation of the AMPK pathway and magnified oxidative stress in esophageal tumor cells [57]. Lv et al. demonstrated that ropivacaine alleviated high-glucose-induced brain microvascular endothelial injury through the downregulation of MMPs and the attenuation of oxidative stress [163]. In breast cancer cells, aspirin-modified metabolites were involved in suppressing lipogenesis, oxidative stress, and neoplastic formation [164]. On the other hand, Kacprzak and Pawliczak reported that aspirin induced oxidative stress in cells, a condition also supported by the inflammatory milieu [165]. In summary, propofol, VAs, opioids, LAs, and NSAIDs may induce uncertain effects on oxidative stress in different cancer processes. In the literature, laboratory research on anesthesia in relation to oxidative stress in PC progression is still limited, and further investigation is needed.

#### 3.7.2. Clinical Studies

In clinical practice, propofol has been proven to have antioxidant capability, and this amelioration of oxidative stress correlates with its enhancement of postoperative prognoses and its association with faster recovery, which is why it is regarded as an ideal anesthetic [157,166]. Ucar et al. reported no significant plasma malondialdehyde changes in 29 liver donors with the use of propofol anesthesia before and after surgery [167]. However, isoflurane and sevoflurane have no influence on oxidative stress, inflammation, and DNA damage in patients undergoing minor surgeries [116]. Purdy et al. reported that the rectus sheath block with levobupivacaine did not significantly affect oxidative stress [168]. To the best of our knowledge, the relationship between anesthesia and oxidative stress on PC progress is still unclear, and further investigation is required.

## 4. The Effects of Anesthetics on PC Progression: From Laboratory to Clinical Studies

### 4.1. Laboratory Studies

#### 4.1.1. Propofol and PC Progression

Several laboratory studies revealed that propofol inhibited PC progression via different pathways [73,74,75,76,77,78,79,80,81]. Chen et al. reported that propofol attenuated PC malignant potential via the inhibition of NMDA receptors [73]. Wang et al. showed that propofol inhibited migration and induced apoptosis of PC cells through *miR-34a*-mediated E-cadherin and LOC285194 signals [75]. In addition, an in vitro experiment by Du et al. demonstrated that propofol suppressed nuclear factor-κB activity and consequently resulted in apoptosis and augmentation of gemcitabine sensitivity in PC cells [76]. Propofol also suppressed PC proliferation and metastasis by means of *miR-328* upregulation and ADAM8 downregulation. [77] Following induction of ADAM8 downregulation, propofol inhibited PC proliferation and migration [74]. Furthermore, propofol repressed ADAM8 both in a hypoxic environment and via SP1 to reduce PC progression. [78,79]. In addition, Wang et al. revealed that propofol depressed PC proliferation and invasion by microRNA-133a upregulation [80]. Liu et al. reported that propofol reduced PC growth and invasion via *miR-21*/Slug signaling regulation [81]. (Table 1, Figure 1 and Figure 2).

#### 4.1.2. VAs and PC Progression

To the best of our knowledge, there is no laboratory study on the relationship between VAs and PC progression, and further research is needed.

#### 4.1.3. Opioids and PC Progression

A recent laboratory study revealed that fentanyl administration decreased the number of cancer and cancer stem cells in PC, decreased the gene expression of stem cell markers, and increased the expression of apoptosis-related genes [86]. (Table 1, Figure 1 and Figure 2).

#### 4.1.4. LAs and PC Progression

In the literature, there are very few laboratory studies on the relationship between LAs and PC progression. In a recent laboratory study, high concentrations of ropivacaine or bupivacaine showed an antiproliferative effect on PC cells [87]. (Table 1, Figure 1 and Figure 2).

#### 4.1.5. NSAIDs and PC Progression

Indometacin ameliorates high-glucose-induced proliferation and invasion by upregulating E-cadherin (EMT) in PC cells [25]. Aspirin counteracts PC stem cell features, desmoplasia, and gemcitabine resistance [82]. In preclinical models of PC, Zhang et al. showed that COX-2 inhibition potentiated the efficacy of VEGF blockade and promoted an immune-stimulatory microenvironment [83]. Perugini et al. demonstrated that sodium salicylate inhibited proliferation and induced G1 cell cycle arrest in human PC cell lines [84]. Indometacin inhibited the proliferation and activation of human pancreatic stellate cells through the downregulation of COX-2 [85] (Table 1, Figure 1 and Figure 2).

#### 4.1.6. Midazolam and PC Progression

The antitumorigenic effects of midazolam are partly related to the peripheral benzodiazepine receptor [169]. Midazolam inhibited cell migrations of human lung carcinoma and neuroglioma, possessed properties of apoptotic induction through the intrinsic mitochondrial pathway, potentially caused mitochondrial membrane decrement, and led to improvement in ROS expression in vitro, which might be attributed to peripheral benzodiazepine receptor activation [169]. In addition, midazolam significantly suppressed proliferation of lung carcinoma in xenograft mice [169]. Oshima et al. reported that midazolam conferred antitumorigenic effects against PC neoplastic growth and achieved local infiltration of tumor-associated neutrophils, macrophages, and polymorphonuclear myeloid-derived suppressor cells [88] (Table 1, Figure 1 and Figure 2).

#### 4.1.7. Ketamine and PC Progression

Malsy et al. reported that ketamine decreased proliferation and reduced expression of NFATc2 in PC cells [89]. They also showed that ketamine reduced proliferation and apoptosis in PC cells [90] (Table 1, Figure 1 and Figure 2).

#### 4.1.8. Tramadol and PC Progression

In a recent study, the antitumor effects of tramadol appeared to involve inhibition of proliferation, induction of apoptosis, and effects on 5-HT2B receptors and TRPV-1 in breast cancer cells [170]. In a xenograft mouse model, Kim et al. showed that a clinical dose of tramadol had antitumor effects on MCF-7-cell-derived breast cancer [171]. In addition, tramadol may regulate EMT and exert cytotoxic effects in breast cancer cells [172]. However, literature on tramadol in PC progression is lacking, and further laboratory studies are required.

#### 4.1.9. Dexmedetomidine (DEX) and PC Progression

In gastrointestinal cancers, DEX promoted the progression of hepatocellular carcinoma through hepatic stellate cell activation [173], but DEX suppressed growth and metastasis while facilitating the apoptosis of esophageal cancer cells by upregulating the abundance of miR-143-3p and reducing the level of EPS8 in vivo and in vitro [174]. To date, laboratory research on DEX in PC progression is lacking, and further investigation is necessary.

### 4.2. Clinical Studies

Recent clinical studies reported that anesthesia may affect cancer outcomes [11,70,175,176]. Data also showed that anesthesia may inhibit PC proliferation and metastasis. Here, we review the clinical effects of anesthetics on PC progression. 

#### 4.2.1. Propofol and PC Progression

In a retrospective propensity score matched cohort, Soliz et al. showed that propofol anesthesia involved risk of low-grade (grade 1 or 2) complications, but carried risk of recurrence, metastasis, or mortality, compared with desflurane anesthesia in PC surgery [10]. Our recent study showed that propofol anesthesia was correlated with enhanced survival (hazard ratio (HR), 0.65; 95% confidence interval (CI), 0.42–0.99; *P* = 0.047) in the matched analysis by comparison with desflurane anesthesia. The subgroup analyses also revealed significant improvement in cancer-specific survival (HR, 0.63; 95% CI, 0.40–0.97; *P* = 0.037) in PC patients undergoing propofol anesthesia. Additionally, fewer postoperative recurrences were noticed in the propofol group than in the desflurane group (HR, 0.55; 95% CI, 0.34–0.90; *P* = 0.028) in the matched analysis [11]. In summary, propofol may inhibit PC progression. Further prospective studies are needed to confirm the effects of propofol on PC progression (Table 1, Figure 1 and Figure 2).

#### 4.2.2. VAs and PC Progression

As mentioned above, desflurane was inferior to propofol anesthesia on patient outcomes after PC surgery [10,11]. The few available studies suggest that VAs may be associated with worse PC progression compared with propofol anesthesia. Further studies are needed to confirm the effects of VAs on PC progression (Table 1, Figure 1 and Figure 2).

#### 4.2.3. Opioids and PC Progression

Steele et al. reported that high opioid use was associated with decreased survival (adjusted HR = 2.76; 95% CI: 1.39–5.48) in newly diagnosed stage IV PC [66]. Abdel-Rahman et al. also reported that opioid use was associated with worse overall survival (HR, 1.245; 95% CI, 1.063–1.459; *P* = 0.007) in patients with PC [67]. Call et al. showed that intraoperative opioid use was not associated with decreased survival in PC surgery [68]. Paradoxically, Zylberberg et al. reported that opioid prescription was associated with increased survival in older adult patients with PC; this might be due to the impact of cancer-related pain [69]. Accordingly, opioids may induce uncertain effects in PC progression. Further investigation is required to determine the effects of opioids on PC progression (Table 1, Figure 1 and Figure 2).

#### 4.2.4. LAs and PC Progression

Cata et al. reported that lidocaine showed a major stimulatory activity on the function of NK cells, and concluded that perioperative lidocaine may diminish the effects of surgery by increasing the cytolytic activity of NK cells [71]. Zhang et al. investigated that significantly higher rates of 1-year and 3-year survival were found in the lidocaine group than in the non-lidocaine group (68.0% vs. 62.6%, *P* < 0.001; 34.1% vs. 27.2%, *P* = 0.011). The multivariable analysis showed that intraoperative administration of lidocaine improved overall survival (HR = 0.616; 95% CI, 0.290–0.783; *P* = 0.013) in patients undergoing pancreatectomy [12]. A clinical study reported that peridural anesthesia with ropivacaine may improve the oncological outcome of PC patients [72]. Chen et al. also reported that intraoperative epidural infusion with a high concentration of ropivacaine was associated with improved overall survival in PC patients undergoing pancreatectomy [70]. However, Lin et al. also showed no significant associations between epidural analgesia with bupivacaine and cancer recurrence and overall survival after curative surgery for PC [177]. Accordingly, LAs in epidural anesthesia (but not analgesia) as well as intravenous lidocaine infusion may induce positive effects on PC progression (Table 1, Figure 1 and Figure 2).

#### 4.2.5. NSAIDs and PC Progression

A meta-analysis study of the research conducted from 2002 to 2017 revealed no significant association between aspirin use and mortality risk in PC [63]. In addition, Risch et al. reported that aspirin use reduced the risk of PC [64]. Recently, Pretzsch et al. demonstrated that aspirin intake proved to be independently associated with improved overall survival, disease-free survival, and hematogenous metastasis-free survival in PC surgery [65]. Accordingly, it can be theorized that NSAIDs may induce positive effects on PC progression. Further investigation is required to verify the effects of different kinds and dosages of NSAIDs on PC progression (Table 1, Figure 1 and Figure 2).

#### 4.2.6. Midazolam and PC Progression

To the best of our knowledge, there are no clinical studies on the relationship between midazolam and PC progression (Table 1, Figure 1 and Figure 2).

#### 4.2.7. Ketamine and PC Progression

There are no clinical studies in the literature regarding the relationship between ketamine and PC progression.

#### 4.2.8. Tramadol and PC Progression

Clinicians may prescribe tramadol for pain management in PC patients [178]. Tramadol promotes or preserves immunity, including NK cell activity, which is important in anticancer defenses [179]. Patients who received tramadol had a decreased risk of postoperative recurrence and mortality following breast cancer surgery [170]. However, the literature on tramadol use in PC is lacking, and further investigation is required to verify its effects on PC progression.

#### 4.2.9. Dexmedetomidine (DEX) and PC Progression

DEX infusion may negatively affect lung cancer surgery outcome [180]. However, DEX may not affect survival in the context of cytoreductive surgery with hyperthermic intraperitoneal chemotherapy for peritoneal carcinomatosis [181]. DEX may induce uncertain effects in the progression of different cancers. The clinical data regarding the effects of DEX on PC are limited.

## 5. Perioperative Anesthesia Management

### 5.1. Enhanced Recovery after Surgery

Agarwal et al. reported that higher compliance with enhanced recovery after surgery (ERAS) protocols significantly reduces the occurrence of major complications, mortality, and postoperative hospital stays in pancreatic cancer resections [182]. Therefore, the implementation of depth of anesthesia monitoring (such as bispectral index), goal-directed fluid therapy with invasive hemodynamic monitoring (such as cardiac output or stroke volume variation), and postoperative nausea and vomiting prophylaxis with propofol use, in accordance with the recommendations of ERAS, were suggested for improving postoperative outcomes [183,184,185]. Using multimodal analgesia with lidocaine infusion, regional nerve blockade, ketamine, and NSAIDs to reduce intraoperative and postoperative opioid consumption is also suggested [183]. 

### 5.2. Blood Transfusion

Perioperative blood transfusion is common in pancreatic surgery. A meta-analysis study reported that patients receiving perioperative blood transfusion had significantly lower 5-year survival after PC surgery [186]. In addition, Kim et al. showed that intraoperative blood transfusion could be considered an independent prognostic factor in resected PC [187]. However, a recent systematic review revealed a weak association between perioperative blood transfusion in patients undergoing pancreatectomy for PC and worsened survival [188]. Accordingly, it can be stated that a perioperative blood transfusion may induce negative effects on PC progression (Figure 2).

### 5.3. Body Temperature

To the best of our knowledge, the effects of temperature on the PC outcome have not been studied. However, hypothermia has long been considered to correlate with major adverse outcomes in different pathophysiological conditions, such as inflammation and cancer. Therefore, active warming should be applied intraoperatively to maintain the patient’s temperature above 36 °C (Figure 2) [183]. 

### 5.4. Glucose Control

Despite the fact that the importance of the optimal glycemic target during the perioperative period is still controversial, blood glucose levels should be maintained as close as possible to the normal range for patient safety [183]. Perioperative hyperglycemia may increase the risk of surgical site infections in general surgery patients [189]. Perioperative prevention of insulin resistance and hypoglycemia is also recommended [183]. Therefore, we believe that controlled fasting and the maintenance of perioperative blood glucose levels may affect the outcome of PC [183].

### 5.5. Obesity

Recently, Zorbas et al. showed that obesity was significantly associated with higher risk of postoperative complications and mortality in patients with body mass index ≥ 40 after pancreatoduodenectomy [38]. Obesity-associated adipose tissue inflammation may play a central role in the development of PC and the promotion of PC growth [32]. Thus, anesthetics with the effect of inhibiting obesity-induced inflammation may improve postoperative outcomes [35]. VAs, ketamine, opioids, propofol, and regional anesthesia may modulate parts of the immune system in patients with obesity [36]. However, there are no studies on the relationship between anesthetics and obesity-induced inflammation in PC progression. Further investigation is necessary.

## 6. Conclusions

PC is a lethal malignancy associated with high mortality, a statistic related to late diagnosis, surgical resection complications, and a shortage of effective therapies. Surgery remains a key component in long-term PC survival. During surgery, anesthesia may alter the tumor microenvironment and even affect cancer progression. Herein, we endeavored to summarize the role of anesthesia in the PC microenvironment and in PC progression. The scarce evidence currently available suggests that propofol, NASIDS, midazolam, ketamine, and LAs may have a positive impact. While the effects of propofol, NASIDS, and LAs have been examined in PC patients, findings regarding midazolam and ketamine are based on cell or animal data, and have not yet been validated in humans. As yet, very little evidence exists to recommend specific anesthetics or analgesic techniques for PC surgery for optimal risk-reduction with regard to tumor recurrence or metastasis. Therefore, additional in-depth research and clinical trials are necessary to explore the molecular mechanisms involved in the relationship between anesthesia and PC microenvironments and progression.

## Figures and Tables

**Figure 1 cancers-14-02684-f001:**
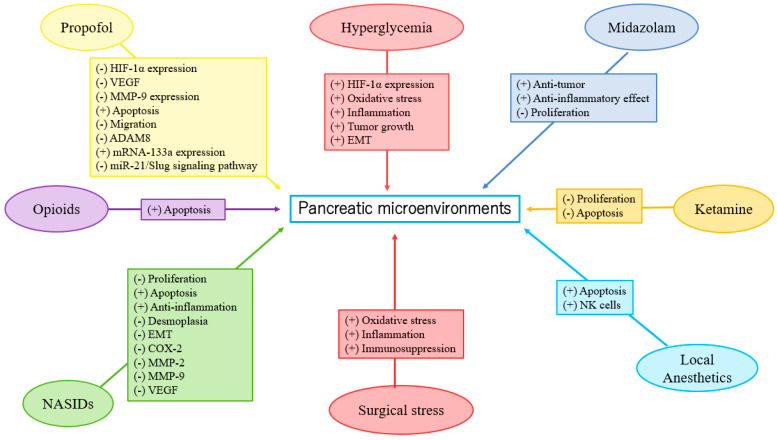
Anesthesia in pancreatic microenvironments.

**Figure 2 cancers-14-02684-f002:**
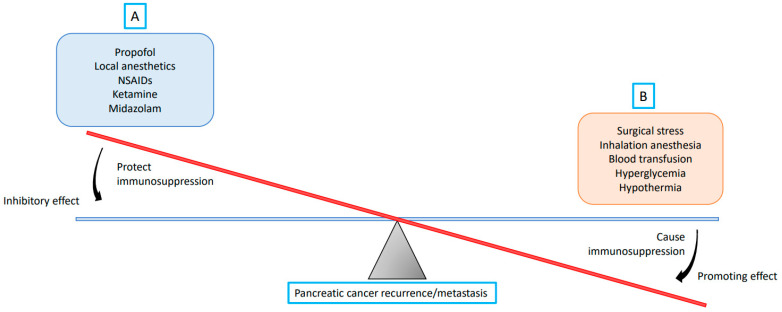
A theory on the balance between recurrence/metastasis-promoting and -inhibiting factors during pancreatic cancer surgery. The strength of the promoting effect relies on the extent of the pancreatic cancer surgery, and the strength of the inhibitory effect originates in the inhibiting factors selected. (**A**) Propofol, local anesthetics, non-steroidal anti-inflammatory drugs (NSAIDs), ketamine, and midazolam reduce pancreatic cancer recurrence/metastasis by protecting against immunosuppression; (**B**) Surgical stress, inhalation anesthesia, blood transfusion, hyperglycemia, and hypothermia promote pancreatic cancer recurrence/metastasis by causing immunosuppression.

**Table 1 cancers-14-02684-t001:** The existing studies on the effects of anesthetics/analgesics on clinical outcomes and pancreatic microenvironments.

Type of Anesthetics/Analgesics	Effects
Clinical studies Propofol/VAs	Propofol was associated with no or low-grade complication compared with desflurane in PC surgery [10]; propofol anesthesia was associated with better survival than desflurane anesthesia in PC surgery [11].
NSAIDs	In a systematic review of observational studies, there was no signification association between aspirin use and mortality risk in PC [63]; aspirin use reduced risk of PC [64]; aspirin was associated with improved overall survival and improved disease-free survival in PC surgery [65].
Opioids	High opioid consumption was related to decreased survival rates in newly diagnosed stage IV PC patients [66]; opioid prescription was associated with poor overall survival among PC patients [67]; there was an insignificant relationship between intraoperative opioid use and decreased survival in PC surgery [68]; administration of opioids was associated with prolonged survival in older adult patients with PC [69].
LAs	Intraoperative administration of intravenous lidocaine was associated with improvement of overall survival in PC patients [12]; intraoperatively epidural ropivacaine infusion was associated with survival improvement in PC patients [70]; perioperative lidocaine administration might be beneficial to the function of NK cells in PC surgery [71]; peridural anesthesia with ropivacaine might improve the oncological outcome of PC patients [72].
Experimental studies Propofol	Propofol attenuated malignant potential by inhibiting HIF-1α and VEGF expression [73]; PC cell growth was inhibited by propofol via suppression of MMP-9 expression [74]; propofol inhibited migration and induced apoptosis [75]; propofol induced apoptosis in PC cells in vitro [76]; propofol inhibited PC progression by downregulating ADAM8 [77,78,79]; propofol suppressed proliferation and invasion of PC cells by upregulating microRNA-133a expression [80]; propofol inhibited growth and invasion of PC cells through regulation of the miR-21/Slug signaling pathway [81].
NSAIDs	Indometacin ameliorated high glucose-induced proliferation and invasion by upregulating E-cadherin (EMT) in PC cells [25]; aspirin counteracted PC stem cell features and desmoplasia and gemcitabine resistance [82]; COX-2 inhibition promoted an immune-stimulatory microenvironment in preclinical models of PC [83]; sodium salicylate inhibited proliferation and induced G1 cell cycle arrest in human PC cell lines [84]; indometacin inhibited proliferation and activation of pancreatic stellate cells through the downregulation of COX-2 [85].
Opioids	Fentanyl decreased gene expression of PC stem cell markers and increased expression of apoptosis-related genes [86].
LAs	High concentrations of ropivacaine or bupivacaine revealed antiproliferative potency in PC cells [87].
Midazolam	Midazolam exhibited antitumor (anti-proliferation) and anti-inflammatory effects in a mouse model of PC [88].
Ketamine	Ketamine significantly inhibited proliferation in PC cells [89]; ketamine significantly inhibited proliferation and apoptosis in PC cells [90].

HIF = hypoxia-inducible factor; Las = local anesthetics; MMP = matrix metalloproteinases; NSAIDs = non-steroidal anti-inflammatory drugs; PC = pancreatic cancer; Vas = volatile anesthetics; VEGF = vascular endothelial growth factor.

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
