# Peer review of "Pancreatic Cancer and Microenvironments: Implications of Anesthesia"

_cancers, 2022, doi:10.3390/cancers14112684_

Round 1

Reviewer 1 Report

Lai et al. present a detailed review on the implications of anesthesia on PDAC progression.

Major remarks

All aspects of anesthesia are examined, the arrangement (hyperglycemia, EMT HIF-1a, MMP-9; inflammation, apoptosis, autophagy, oxidative stress; anesthesia; anesthetic adjuvants; blood transfusions, body temperature) appears to be a bit chaotic. E.g. I do not understand why you explain different anesthetics and analgetics in chapter 4 and have another chapter 5 with ketamine? Eventually you should rearrange your manuscript (e.g. tumor factors; drugs, perioperative management).

The advantage of the manuscript is the detail, in which the different topics are discussed with their impact on PDAC progression. But clinical data (patients) are mixed with laboratory data (cell lines), often in the same paragraph, and this is the main drawback of the paper: the clincal impact is not clear. An effect shown in cell lines does not necessarily translate into a clincal impact. This must be strictly differentiated. Consider a paragraph in which you state which of the findings have already been examined on PDAC patients and which are cell or animal data, not yet validated in man.

Minor remarks

Check for misspellings (e.g. line 169)

Reviewer 2 Report

This review is well organized and easy to read. Focusing on surgical anesthesia in the progression of the PC is unfamiliar to cancer biology. Therefore, this is very valuable for us. This version is fine for publishing, but this reviewer recommends the following changes as minor comments.

minor comments

  1. For #1, and 6, pay attention to choosing more appropriate references according to the sentence.
  2. Check for a typo on line 169.

Reviewer 3 Report

Manuscript (Numbered 1710834) titled: “Pancreatic Cancer and Microenvironments: Implications of Anesthesia”, has been carefully reviewed. The authors provided a narrative review of the effect of anesthesia on recurrence and metastasis of pancreatic cancer. This review is comprehensive and the manuscript is well written. I have only two miner comments.

  1. In line 182, the authors concluded that Propofol might reduce HIF-1α expression on PC. However, this issue needs to expand by providing more evidences before jumping to the conclusion. Recent study by Yue et al. demonstrated that HIF-1α positively regulated miR-212 expression and resulted in pancreatic ductal adenocarcinoma progression. (Yue H, Liu L, Song Z. miR-212 regulated by HIF-1α promotes the progression of pancreatic cancer. Exp Ther Med. 2019;17(3):2359-2365. doi:10.3892/etm.2019.7213). Moreover, in a systemic review and meta-analysis, Raji et al. found that miR-212 could be a novel potential biomarker in cancer diagnosis and prognosis. (Raji S, Sahranavard M, Mottaghi M, Sahebkar A. MiR-212 value in prognosis and diagnosis of cancer and its association with patient characteristics: a systematic review and meta-analysis. Cancer Cell Int. 2022 Apr 26;22(1):163. doi: 10.1186/s12935-022-02584-0. PMID: 35473623; PMCID: PMC9044851.). Suggest provide more information regard to the effect of Propofol and other anesthetics on HIF-1α and miR-212 expression.

2. Obesity is associated with higher risk of postoperative complications in patients undergoing pancreatoduodenectomy (PD) and patients with BMI≥40 have increased risk of mortality after PD. (Zorbas K, Wu J, Reddy S, Esnaola N, Karachristos A. Obesity affects outcomes of pancreatoduodenectomy. Pancreatology. 2021 Jun;21(4):824-832. doi: 10.1016/j.pan.2021.02.019. Epub 2021 Mar 4. PMID: 33752975.) If possible, please discuss obesity and PC microenvironments and the possible effect of anesthesia.
